# Improved Bladder Tumor RNA Isolation from Archived Tissues Using Methylene Blue for Normalization, Multiplex RNA Hybridization, Sequencing and Subtyping

**DOI:** 10.3390/ijms231810267

**Published:** 2022-09-06

**Authors:** Stefanie A. Köhler, Lisa Brandl, Pamela L. Strissel, Laura Gloßner, Arif B. Ekici, Miriam Angeloni, Fulvia Ferrazzi, Veronika Bahlinger, Arndt Hartmann, Matthias W. Beckmann, Markus Eckstein, Reiner Strick

**Affiliations:** 1Laboratory for Molecular Medicine, Department of Gynecology and Obstetrics, University Hospital Erlangen, Friedrich-Alexander Universität Erlangen-Nürnberg, Universitaetsstrasse 21-23, 91054 Erlangen, Germany; 2Comprehensive Cancer Center Erlangen-EMN (CCC ER-EMN), University Hospital Erlangen, Östliche Stadtmauerstrasse 30, 91054 Erlangen, Germany; 3Institute of Pathology, University Hospital Erlangen, Friedrich-Alexander Universität Erlangen-Nürnberg, Krankenhausstrasse 8-10, 91054 Erlangen, Germany; 4Institute of Human Genetics, University Hospital Erlangen, Friedrich-Alexander Universität Erlangen-Nürnberg, 91054 Erlangen, Germany; 5Department of Nephropathology, Institute of Pathology, University Hospital Erlangen, Friedrich-Alexander Universität Erlangen-Nürnberg, Krankenhausstrasse 8-10, 91054 Erlangen, Germany

**Keywords:** muscle invasive bladder cancer, FFPE and fresh-frozen tissues, methylene blue, RNA isolation, RNA normalization, 5SrRNA and 5.8SrRNA, multiplex RNA hybridization, luminal and basal subtypes

## Abstract

Methylene blue (MB) is a dye used for histology with clinical importance and intercalates into nucleic acids. After MB staining of formalin fixed paraffin embedded (FFPE) muscle invasive bladder cancer (MIBC) and normal urothelium, specific regions could be microdissected. It is not known if MB influences RNA used for gene expression studies. Therefore, we analyzed MIBC using five different RNA isolation methods comparing patient matched FFPE and fresh frozen (FF) tissues pre-stained with or without MB. We demonstrate a positive impact of MB on RNA integrity with FF tissues using real time PCR with no interference of its chemical properties. FFPE tissues showed no improvement of RNA integrity, which we propose is due to formalin induced nucleotide crosslinks. Using direct multiplex RNA hybridization the best genes for normalization of MIBC and control tissues were identified from 34 reference genes. In addition, 5SrRNA and 5.8SrRNA were distinctive reference genes detecting <200 bp fragments important for mRNA analyses. Using these normalized RNAs from MB stained MIBC and applying multiplex RNA hybridization and mRNA sequencing, a minimal gene expression panel precisely identified luminal and basal MIBC tumor subtypes, important for diagnosis, prognosis and chemotherapy response.

## 1. Introduction

Although the molecular distinction between fractionated RNA and DNA was successful in the beginning of the 1900’s [1] by detection of Uracil instead of Thymine, respectively, bulk isolations of ribonucleic acids date back to the 1940s. These early fractionation studies used acid and alcohol extractions to isolate plant viral RNA [2], or acid with NaCl and ethanol for nucleic acid recovery from bovine pancreas [3]. However, at that time it was not known if the isolated RNA was a single component or a mixture. The establishment of more quantitative and qualitative methods for RNA isolation included new substances like 2 M guanidinium chloride [4], alkaline extraction without alcohol [5], phenol-water extractions [6] and sodium chloride precipitations [7]. Especially, the finding that 4 M guanidinium chloride inhibited pancreatic RNAses [8] resulted in high quality RNA isolation. Furthermore, using chromatographic substances revolutionized the isolation of RNA and DNA becoming the foundation for RNA/DNA binding column purifications. For example, the first columns consisted of methylated serum albumin kieselguhr [9]. Even more superior, protamine-coated kieselguhr [10] columns along with high salt elution demonstrated high quality total RNA and DNA from rat livers, bacteriophages, bacteria and yeast but without RNA or DNA specificities.

The methylene blue (MB) dye, a phenothiazine derivate, was synthesized in the middle of the 19th century and produces a blue color when solubilized in water. Due to its small molecular weight, fast diffusion properties and affinity for nucleic acids, MB was initially used to detect tuberculosis in lung tissues [11] and it is still clinically used today for sentinel node mapping, detection of methemoglobinemia, septic shock and neuroendocrine tumors like insulinomas [12]. In addition, MB, which is United States Food and Drug Administration (FDA) approved, exhibits antiseptic action and can protect tissues from anti-inflammatory and anti-oxidant effects [12]. Importantly, MB also showed chemotherapeutical properties as an antimalarial drug [12,13] and against cancers where MB is used for photodynamic therapy [14,15,16]. The specific ability of MB to stain tissues instated MB as a pathological dye and substituted May–Grünwald–Giemsa [17]. It was also shown that MB specifically binds ssDNA and intercalates dsDNA [18]. MB binds tRNA [19] as well as cellular RNA, where it was used for staining electrophoresed RNA in polyacrylamide gels [20]. Furthermore, MB binds dsRNA and perturbates the dsRNA conformation [21]. However, it was noted that pre-staining 4S and 5S RNAs with MB did not change the migration behavior compared to unstained RNA [22].

For decades, tumor tissues archived via formalin fixed and paraffin embedded (FFPE), fresh or flash-frozen (FF) methodologies represented the main pillars for clinical, therapeutic, translational and basic research. Both ways of archiving tumor tissues have pros and cons, but for RNA and DNA sequencing FF tissues are superior regarding RNA quality, although a few laboratories did not find major quality differences compared to FFPE tissues [23]. Research is ongoing to improve existing methodologies for isolation of RNA especially from FFPE tissues. We previously observed that when MB was incubated with FFPE tissue sections following deparaffinization and rehydration, tumor and normal urothelium regions could easily be identified and microdissected. Therefore, we considered MB as a positive aid for identification and microdissection of specific tissue regions. However, it is not known if MB influences RNA throughout isolation used for gene expression studies.

In the present report, we systematically analyzed muscle invasive bladder cancer (MIBC) tissues using five different RNA isolation methods comparing patient matched FFPE and Tissue-Tek^®^ FF tissues pre-stained with or without MB. These isolation methods were based upon magnetic beads, columns or organic solvent extraction. MIBC is a grave form of bladder cancer arising from urothelium, where patients have a five-year survival of 40–60% [24]. Over the past decade, scientific efforts have led to the discovery of multiple molecular MIBC subtypes, which on the one hand reflect the general biology of MIBC and on the other hand strongly correlate with patient survival, response to chemotherapy and immunotherapy [25]. Protein immunohistochemistry (IHC) is considered the gold standard for identifying the two main MIBC luminal and basal subtypes [26]. Further research has led to a common consensus for tumor subtyping, identifying six biologically and clinically distinct subtypes, which can only be determined by RNA transcriptome expression analysis [25]. Since clinical diagnosis uses FFPE material, only limited amounts are available to isolate and perform RNA expression analyses such as real time PCR (qPCR) or next generation sequencing (NGS) mRNA-seq. Thus, for assessment of MIBC subtypes it is important to isolate high quality RNA and use the most suitable genes for normalization of expression.

Therefore, our main goal in this investigation was to determine if MB impacts RNA during fractionation from MIBC tissues and influences RNA integrity based upon analyses of ubiquitously expressed reference genes, such as *18SrRNA,* with the use of *5SrRNA* and *5.8SrRNA* genes as a new approach for real time qPCR. Considering the recommendation that rRNAs, which constitute 85–90 % of total cellular RNA are highly useful and accurate internal controls and more reliable than other reference genes, a normalization of the isolated RNA to total RNA using especially fluorescent RNA stains is advised [27].

According to multiple publications we chose *18SrRNA* for our RNA integrity analyses due to its high transcript abundance in total RNA, since it was reported to show increased expression rates and reduced degradation compared to other reference genes [28]. It is very important to establish reference genes, which do not show high variabilities between single tumors and control tissues (according to the Minimum Information for Publication of Quantitative Real-time PCR Experiments (MIQE) guidelines [29]); we therefore tested 34 widely used published reference genes with MIBC and control tissues and determined the best genes for normalization in qPCRs and direct multiplex RNA hybridization. As “proof of concept” using microdissection of MB stained FFPE tissues (*n* = 28) and one of the best RNA isolation methods determined in this investigation, multiplex RNA hybridization and mRNA sequencing confirmed luminal and basal MIBC tumor subtypes important for MIBC diagnosis, patient prognosis and chemotherapy [25]. Our results highlight the importance of MB for tissue microdissection enriching target RNA, but MB also has a positive effect on RNA integrity throughout isolation especially using FF tissues.

## 2. Results

Proper handling and chemical processing of tumor tissues as a daily event at Pathology Institutes is absolutely crucial for diagnosis. The retrieval of high quality nucleic acids, such as RNA from archival tissues, is essential for studying the intrinsic gene expression of tumors and can lead to the discovery of new biomarkers as well as profiling tumor subtypes such as MIBC basal and luminal for patient prognosis [25,30]. Present methodologies for analyzing gene expression used in laboratories world-wide are qPCR, RNA hybridization and mRNA sequencing and ultimately bring knowledge into the public domain. To reach high methodological standards for nucleic acid isolation, characterization and normalization by establishing reference genes per tissue type should be continually optimized. It is known that archival FFPE tissues yield more fragmented RNA due to storage time and conditions but especially via chemical modifications by formalin, a solution which is saturated formaldehyde in methanol and water [31]. Since FFPE isolated RNA in general yields smaller fragments, we first sought to test whether MB, which specifically binds nucleic acids, could influence the integrity of RNA using common isolation kits designed for archival tissues. The same methodologies were performed with patient matched archival FF tissue samples for comparison.

### 2.1. MB Tissue Treatment Identifies Histological Regions for RNA Microdissection

MIBC tissues were stained with MB and then specific targeted tumor regions, microdissected for RNA isolation. MB staining was similar to the state of the art hematoxylin and eosin (H&E) regarding the histological appearance (Figure 1).

### 2.2. General Characteristics of Different RNA Isolation Methods for FFPE and FF Tissues Treated with or without MB

For comparison, five different isolation kits were used to extract total RNA from FFPE and FF tissues stained with or without MB (Table 1). Each methodology is described in detail in the Materials and Methods. According to the manufacturer protocols, 5–10 µm tissue sections can be extracted for RNA using methodologies based upon magnetic beads (Methods A, B, D), column chromatography (Method C) or via an organic solvent (Method E). The cellulose coated paramagnetic beads in Method A are described as having a higher binding capacity and provide an isolation with a cleaner eluate than traditional DNA purification. For Method B, silica coated magnetic beads are used to bind nucleic acids. Germanium coated beads used in Method D guarantee an efficient binding of nucleic acids and a higher yield compared to other suppliers. During column chromatography with Method C, RNA together with chaotropic salts binds to the surface of glass fibers in a filter tube. After DNase I treatment and washing steps, the RNA is eluted by a low salt buffer. Lastly, isolation containing an organic solvent as in Method E facilitates the separation of RNA, DNA and proteins. Additionally, differences exist between the methods regarding handling of the samples. For example, for FFPE tissues proteinase K treatment was added prior to addition of a lysis buffer (Method A) or directly to the lysis buffer (Methods B–D) with a 40–60 min incubation (Methods A, B and D) or overnight (Method C). All FF tissues were incubated overnight with proteinase K regardless of the method. Furthermore, all samples were heat treated at 80 °C either before (Methods A and D) or after proteinase K treatment (Methods B and C). Methods A and E did not contain a DNAse I treatment, and thus it was added after the initial procedure. One final difference is the automated purification of RNA for Methods A and B. Within an instrumental run time of 40 min the RNA is isolated, eluted, and purified. This requires purchasing a specific instrument resulting in higher costs per reaction compared to the column based Method C, but might reduce human errors and enable large-scale projects. Methods A and B have a total hands-on time of 3 to 3.5 h, respectively, but only 2.5 h for Method C (Table 1).

### 2.3. MB Tissue Treatment Shows Advantages during RNA Isolation of MIBC Tissues Compared to Other Dyes with Different Methodologies

An initial comparison of extracted RNA from two MIBC FFPE tissues using Method A in the presence of MB revealed statistically significant improved *18SrRNA* values compared to no MB treatment using qPCR (Figure 2A). The MB molecular structure is an organic chloride salt with 3,7-bis(dimethylamino)phenothiazin-5-ium as the counterion (C_16_H_18_ClN_3_S). With this in mind, we asked the question, if other dyes with similar molecular structures or nucleic acid binding properties like MB could also significantly affect *18SrRNA* changes using the same tissues. For example, Tolouidine blue (TB), an organic chloride salt with 3-amino-7-(dimethylamino)-2-methylphenothiazin-5-ium as the counterion (C_15_H_16_ClN_3_S), also led to improved *18SrRNA* values using qPCR. However, Crystal violet (CV), a hexamethylpararosaniline chloride (C_25_H_30_ClN_3_) led to variable *18SrRNA* results (Figure 2A). This supports the suggestion that the common chemical structures of both MB and TB are important for improving RNA isolation Method A.

Upon finding MBs’ positive effect improving *18SrRNA* values of two FFPE MIBC tissues using qPCR, we tested four additional archival MIBCs with other RNA isolation methods (Methods A–D for FFPE; Methods A–E for FF; *n* = 6 total tumors). Results confirmed our initial findings with additional FFPE tissues and Method A (Figure 2B). In contrast, using three other RNA isolation techniques (Methods B–D) with the same six FFPE tissues did not significantly influence RNA integrity in terms of *18SrRNA* using qPCR, supporting no clear benefit of MB. On the other hand, when extracting RNA using the same isolation methods (Methods A–D) with patient matched FF MIBC tissues, a profound statistically significant improvement of *18SrRNA* was found for three of four methods (Methods A, B and C), with MB treatment compared to no MB (Figure 2C). Interestingly, the RNA isolation of FF MIBC using Method D resulted in improved *18SrRNA* without MB. Lastly, Method E (Trizol) showed no significant differences of *18SrRNA* between FF tissues treated with or without MB. Trizol contains phenol and guanidinium thiocyanate and is recommended for extracting FF tissues, although the process is time consuming and phenol contamination is possible. Importantly, we observed that MB during RNA isolation fractionated into the organic phase, and thus was released from nucleic acids in the aqueous phase, supporting no sustained RNA MB interaction. Although *18SrRNA* is a well-known reference gene, importantly, we normalized to the total RNA based on several quantification methods including a fluorescent RNA stain detecting down to 5 ng/mL RNA [32], and additionally normalized *18SrRNA* to another reference gene (*5.8SrRNA*) to calculate ΔCts. Analysing ΔCts of the RNAs isolated from the different methods with MB showed that the highly significant changes (A _+MB and C_+MB) did not change in significance (Figure 2E). However, the lower significant changes using *18SrRNA* Ct (A_+MB FFPE, B_+MB FF and D_−MB FF) were no longer significant (Figure 2B–E).

In summary, our results show that, in the presence of MB, RNA isolation from FFPE tissues using Method A and from FF tissues using Methods A–C, statistically significantly improved RNA integrity using qPCR for *18SrRNA* Ct, and especially for Methods A and C from FF tissues even after normalization.

In order to rule out any artifactual effects due to MB, which has an emission at 610 and 664 nm [33], we performed several control experiments. Due to the fact that MB interacts with nucleic acids, we asked whether MB could interfere with optical density readings affecting qPCR. Analyzing optical density measurements of isolated RNA from MIBC tissues from the different isolation methods, in the presence or absence of MB, showed an RNA absorbance at 260 nm, but not at 610 nm and 664 nm. Secondly, if MB was bound to RNA but below the level of detection, we tested the possibility, if MB could interfere with the qPCR results using SYBR-Green, which binds DNA with an absorbance of ~497 nm and emission of ~520 nm. We added MB in dilutions from 1:10 to 1:2000 to purified RNA and then performed cDNA-synthesis. In addition, MB was added to synthesized cDNA and then employed qPCR for *18SrRNA*. Results showed for all experiments, except for the highest concentration of a 1:10 MB dilution with cDNA, no detectable differences in amplification signals in the presence or absence of MB (Appendix A). These results suggest that MB based upon its emission properties does not interfere with qPCR causing artifacts.

### 2.4. Evaluating the Integrity of Isolated RNA from FFPE and FF Tumor Tissues

In order to determine RNA integrity, we analyzed the RNA integrity number (RIN), which evaluates *18SrRNA* (1869 bp; e.g., NR_003286) and *28SrRNA* (5070 bp; e.g., NR_003287) regions compared to the total area under the curve, including the height of the *28SrRNA* peak [34]. Analyzing RNA from two patient matched FFPE and FF MIBC tissues (#126, #128) using all five isolation methods, with or without MB, RINs ranged between 2.2 and 3.5, without any significant improvement using MB (Table 1, Appendix A–E). Due to the findings that archival tissues mostly result in inadequate RINs, we tested the same RNAs as above using a novel RNA integrity analysis called DV200 [35]. The DV200 determines the percentage of RNA fragments over 200 nucleotides as developed by Agilent (http://urx.red/OB4Y) (accessed on 8 October 2021) [35]. However, no significant differences were found comparing FFPE and FF using the different isolation methods in the presence or absence of MB (Table 1, Appendix A–E).

In order to further address the integrity of the different isolated RNAs, we implemented a new reference approach to detect RNA in the amplification range of approximately ~100 bp for qPCRs. Routinely, qPCR primers for reference and target genes are designed to amplify regions between 90 and 150 bp. Uniquely, our designed primers detected the constitutively expressed *5SrRNA* (121 bp; e.g., X51545.1) and *5.8SrRNA* (156 bp; e.g., NR_146120.1). Like *18SrRNA*, *5.8SrRNA* is transcribed by RNA-Polymerase I, whereas *5SrRNA* is transcribed by RNA-Polymerase III [36]. In contrast to all other reference and target genes, the advantage of assessing *5S* and *5.8SrRNAs* using our primer design shows that amplification for each gene can only occur, if these rRNAs are full length fragments. Therefore, amplification of both rRNAs could reflect a quality qPCR reference for smaller RNAs below 200bp. QPCR results showed that the addition of MB significantly improved *5S* and *5.8SrRNA* expression similar to *18SrRNA* (Figure 3A,B). In addition to implementing both rRNAs as reference genes, we also examined the ratios of *5SrRNA* to *5.8SrRNA* gene expression of FFPE and FF MIBC tissues for all RNA isolation methods and from five normal placenta tissues (Figure 3C). Interestingly, the FFPE *5SrRNA* to *5.8SrRNA* ratios (1.01 to 1.28) were generally higher compared to FF tissues (0.94 to 1.23) and human placenta (1.07). Especially for FF tissue isolation with Method C, MB improved the overall ratios (+MB: 0.98–1.13; −MB: 0.63–0.95).

### 2.5. GUSB, TBP and UBC Are the Most Suited Reference Genes for MIBC and Normal Bladder Urothelium

In contrast to qPCR, direct multiplex RNA hybridization is an amplification free, highly multiplexed and throughput technique with significant sensitivity and robustness [37]. In order to determine the most suitable reference genes for MIBC and normal urothelium, 34 commonly used reference genes from the literature were tested using direct multiplex RNA hybridization and FFPE tissues (Appendix A). We chose Method B for RNA isolation, which showed excellent *18SrRNA*, *5SrRNA* and *5.8SrRNA* values in the presence of MB prior treatment (Figure 2B). In addition to the 6 FFPE MIBC tissues we expanded our cohort to a total of 28 MIBC and 5 normal urothelium isolated RNAs also using Method B and MB for microdissection. All raw counts were first normalized to the positive plate control to minimize technical errors between runs. Subsequently, all expression counts of the MIBC and normal urothelium were analyzed for each reference gene by calculating the coefficient of variation (COV) or relative variability, which represents the standard deviation of a group of values divided by their mean. The COV was determined in percent with the lowest COV supporting the best constant reference gene expression (Figure 4A,B). Analysis revealed three reference genes: Glucuronidase beta (*GUSB*; HGNC:4696) (COV 61.44%), Ubiquitin C (*UBC*; HGNC:12468) (COV 61.38%) and TATA-box binding protein (*TBP*; HGNC:11588) (COV 70.99%). *GUSB* is involved in the degradation of glycosaminoglycans [38]. The TATA box-binding protein (*TBP*) is the DNA-binding subunit of RNA polymerase II transcription factor D, essential for expression of most protein-encoding genes [39]. As a member of the ubiquitin family, Ubiquitin C (*UBC*) is a polyubiquitin precursor, and conjugation of ubiquitin is associated with many cell functions, especially with protein degradation [40]. Finally, these genes were also positively confirmed using qPCR, as indicated by slope values of standard curves for *18SrRNA* −3.221 (R^2^: 0.997), for *TBP* −3.55 (R^2^: 0.990) and for *UBC* −3.69 (R^2^: 0.979) (Appendix A–H). Interestingly, comparing the standard curves, slopes and R^2^ of RNA isolated from two different MIBC FF tissues further indicates that the isolation technique results in similar RNA concentrations and degradations and confirms highly similar amplification efficiencies and linearity.

### 2.6. MIBC Subtyping of Luminal and Basal Tumors Comparing Protein IHC, Direct Multiplex RNA Hybridization and mRNA Sequencing

MIBC can be subtyped into clinically significant luminal and basal subtypes correlating with patient prognosis and response to chemotherapy [25]. This 2020 published consensus RNA transcriptome subtyping of 1750 MIBC defined six biologically different tumor subtypes: three luminal subtypes (luminal papillary/LumP, luminal genomically unstable/LumU and luminal/Lum); one stroma-rich subtype, with marked stroma signatures, where most tumors classify with luminal; one basal/squamous subtype and one subtype with neuroendocrine differentiation (NE-like). To validate our above results in terms of clinical relevance, we implemented the same RNA processes as above (Method B for RNA isolation with MB prior tissue treatment) and compared the performance of direct multiplex RNA hybridization subtyping (normalized as above) with two objective reference standards: protein IHC subtyping and mRNA consensus subtyping to test for congruence. Using unsupervised hierarchical clustering of six subtyping genes assessed by multiplex RNA hybridization (basal differentiation: *KRT5*, *KRT14*, *CD44*; luminal differentiation: *KRT20*, *GATA3*, and *FOXA1*), 28 tumor samples were each classified as either basal or luminal subtypes (Figure 4C). The concordance rate with protein IHC and simplified mRNA reference subtyping was 100% (Figure 4D). Furthermore, all additional luminal subsets of the consensus subtyping were classified as luminal using the direct multiplex RNA hybridization subtyping panel (Figure 4D).

## 3. Discussion

Routine pathological tissue processing to establish biobanks of archival tissues generally involves FFPE, Tissue-Tek^®^ or liquid nitrogen. All tissue processing methods share one critical step, the time period after surgery prior to fixation. This waiting time varies between patient surgeries and understandably cannot be standardized. Following tissue oxygen ablation, rapid RNA fragmentation begins and progresses over time as more anoxic as well as hypoxic conditions due to a reduction of pH prevail [41]. Methods involving immediate freezing stabilize RNA and DNA structure; however, it is well known that formalin preserves RNA and DNA structures by introducing nucleotide crosslinking, especially with proteins, but also results in strand breakage into fragments with an average of 100–200 bp length [41,42,43,44]. Furthermore, formalin fixed tissues can lead to additional RNA fragmentation following paraffin embedding upon exposure to light. On the other hand, paraffin can lead to high molecular weight RNA aggregates, resulting in low yield and poor quality RNA [45]. Thus, a limiting factor in retrieving RNA from FFPE tissues is complete paraffin removal, rehydration and protein breakdown to help free nucleotide crosslinks [41,46]. Even more problematic is the additional RNA damage during fractionation that can occur with the use of various isolation methods. Therefore, the task of isolating RNA from a preserved tissue state is not only challenging as regards choice of the correct method, but also further research is needed to improve RNA integrity.

Chemical dyes for the histology of pathological tissues are critical for diagnosis, but can also be an aid for microdissection and enrichment of specific tissue regions for isolation of RNA and DNA needed for molecular studies. To date no systematic studies comparing FFPE with FF human tissues treated with or without MB and addressing its influence on RNA integrity had been performed. Thus, we investigated whether MB staining of FFPE and FF tissues has an influence on RNA integrity, using four or five isolation methods, respectively. Our study brings forth new knowledge that MB tissue staining prior to RNA fractionation results in improved RNA integrity especially with FF tissues as determined by qPCR. Importantly, we found that MB tissue staining is easily done within 20 s and its emission does not influence RNA optical density measurements or qPCR using SYBR-Green. As both MB and toluidine blue share similar molecular structures and significantly improved *18SrRNA* values using qPCR, MB led to an overall improvement of RNA integrity primarily with FF tissues, which was also reflected by qPCR of *5SrRNA* and *5.8SrRNA* (Figure 2 and Figure 3). Except for isolation Method A, all other methodologies led to no significant *18SrRNA*, *5SrRNA* or *5.8SrRNA* differences when FFPE tissues were treated with or without MB.

Examining similarities and differences of RNA extracted from FF and FFPE patient matched MIBC tissues treated with or without MB helps to explain the molecular basis of how MB may positively influence RNA integrity. Although FF and FFPE fractionated RNA fragments were generally smaller in size, no significant RIN or DV200 differences were found. Using our new approach of using *5SrRNA* and *5.8SrRNA* as reference genes detecting less than 200 bp, both genes showed improvement using qPCR from FF tissues treated with MB, but not with FFPE tissues. This not only confirms our *18SrRNA* results for both FF and FFPE tissues, but also supports increased preservation of RNA fragments throughout isolation from FF tissues treated with MB. We propose that the overall improvement of RNA integrity, when implementing MB with FF tissues, most likely stems from the lack of formalin and paraffin fixation and crosslinking found for FFPE tissues [41,46,47,48]. With the same time period prior to fixation, RNA from patient matched FFPE and FF tissues without MB treatment, shows a negative effect of formalin reducing RNA integrity (FFPE Ct mean = 17.71; FF Ct mean = 12.88 for *18SrRNA*; *n* = 6 tumors; isolation Methods A–D). In addition, use of MB also showed a statistically significant benefit to RNA integrity when comparing individual isolation methods for FF tissues (e.g., Method C: no MB, Ct mean = 13.99 and with MB Ct mean = 10.04 for *18SrRNA*) (Figure 2 and Figure 3).

Since RNA fragmentation is ongoing in tissues prior to fixation and upon formalin treatment when nucleic acids cross link, specifically RNA, this leads to hydrolysis [41,47,48]. Therefore, we support the idea that formalin fixed RNA fragments are preserved in structure, but during isolation the RNA is more resistant to MB in stabilizing its structure and thus the RNA becomes more fragile leading to poorer integrity. This could be due to remaining cross linked residues and residual paraffin explaining why no differences in *18SrRNA*, *5SrRNA* and *5.8SrRNA* qPCR were observed for most isolation methods using FFPE tissues with or without MB. On the other hand, we propose that MB RNA intercalation in FF tissues stabilizes and preserves RNA structure from further fragmentation during isolation resulting in a benefit for RNA integrity. In addition to using *5SrRNA* and *5.8SrRNA* as marker genes for the integrity of smaller RNA fragments below 200 bp, expression of *5SrRNA* (via RNA polymerase III) and *5.8SrRNA* (via RNA polymerase I) can also represent an advanced quality check of the cellular state. For example, *Tetrahymena* cells showed 1:1 molar ratios of *5SrRNA*: *5.8SrRNA* during log growth phase. In contrast, starved *Tetrahymena* cells synthesized approximately 15% more *5SrRNA* over *5.8SrRNA*, supporting *5SrRNAs* as a biomarker for changes in physiological cellular conditions [49]. Examining *5SrRNA*:*5.8SrRNA* gene expression ratios for all RNA isolation methods, FF MIBC tissues demonstrated ratios closer to 1.0, which could reflect more stable conditions with less cellular stress following loss of blood supply prior to fixation for these six bladder tumors (Figure 3C). Alternatively, *5SrRNA*:*5.8SrRNA* gene expression ratios were higher than 1.0 for matched tumor FFPE tissues with identical pre-fixation waiting times, suggesting a higher stability of *5SrRNA* during formalin induced RNA fragmentation. Note that, for example, *5SrRNA* is used as a preferential reference RNA for micro RNA (miRNA) identification in body fluids [50].

Another important advantage of MB is the similar staining of tissues compared to H&E (Figure 1). The relative short staining protocol of MB versus H&E not only has a benefit of less time, but probably also a positive effect for RNA quality. Although we did not compare RNA integrity of MB versus H&E stained tissues, it was shown previously that RNA isolated from H&E stained tissues harbors the risk of RNA damages, especially without RNAse inhibitors [51]. The risk of RNA degradation was also found to be dependent on the time of the H&E procedure, because a reduction from 26 min to 5 min for the H&E staining improved the RNA quality [52]. Thus, MB is similar to H&E staining for histology and the advantage of MB per se for RNA quality could indicate MB as a superior stain for laser capture microdissection or spatial sequencing.

Several publications described the identification of reference genes for bladder cancer [53,54,55]. For example, using 9 candidate genes and 14 tumor and control tissues, Ohl et al. found SDH and *TBP* as most stable genes using qPCR (Appendix A) [53]. Concerning *TBP* as a suitable reference gene, this was also established in this present investigation using direct multiplex RNA hybridization. Interestingly, we also identified *GUSB* and *TBP* as suitable references for the MIBC and control bladder. Considering the additional steps of cDNA synthesis and amplifications, several factors can mask the best reference genes as also mentioned earlier, such as primers and normalization [28]; thus one recommendation could be to analyze reference genes of tissues with direct multiplex RNA hybridizations and confirm using qPCR.

MIBC is subtyped into clinically relevant luminal and basal subtypes important for patient prognosis, chemotherapy and immunotherapy response [25]. This is of particular importance, since MIBC subtype assessment is likely to become a routine clinical standard in the near future, if current ongoing clinical trials validate the importance of subtypes prospectively in randomized settings [25]. Currently, several protein or RNA methodologies exist to assess luminal basal molecular subtypes, e.g., protein subtyping using IHC [56] or mRNA consensus subtyping [25]. Our study demonstrates that when using MB to enrich tumor regions for RNA isolation, along with gene expression normalization using high quality reference genes, high concordance rates between IHC and different gene expression tumor subtyping approaches is achieved. Previously, we implemented a larger panel of genes for subtyping both luminal and basal MIBC subtypes using RNA hybridization [30]. In this report, using a minimum of 3 genes to identify each tumor subtype with both RNA methodologies demonstrated a high specificity. Therefore, it is reasonable that microdissection of targeted tissue regions stained with MB leads to an enriched tumor RNA population with a greater specificity of single copy genes. Due to the clinical significance of MIBC subtyping for patient prognosis and therapy response, our findings clearly underline the importance of proper handling of FFPE tissues. These include standardized RNA isolation methods resulting in high integrity as well as use of correct reference genes in order to achieve precise subtyping results using IHC subtyping and/or direct mRNA hybridization or NGS mRNA-seq. This is relevant as usually only FFPE tissue samples are available in routine patient care scenarios.

## 4. Materials and Methods

### 4.1. Ethics Approval and Consent to Participate

The present study was approved by the Ethic Commission of the Friedrich-Alexander-University of Erlangen-Nürnberg (252_20 B; 302_19 Bc; 353_15 B). All the procedures were performed in accordance with the ethical standards established in the 1964 Declaration of Helsinki and later amendments. All patients gave written consent for the use of their tumor or normal tissue material.

### 4.2. Formalin-Fixed, Fresh-Frozen and Flash-Frozen Tissue Samples

Formalin-fixed, paraffin-embedded (FFPE) and Tissue-Tek^®^ embedded fresh-frozen (FF) archived tissues (2008–2019) included in this study were: (1) FFPE MIBC (*n* = 6) and matched FF MIBC samples (*n* = 6); FFPE MIBC samples (*n* = 23); (2) FFPE normal control urothelium (*n* = 5) from non-cancer patients. In addition, five control placental tissues (3rd trimester) were flash-frozen into liquid nitrogen.

### 4.3. Tissue Processing and Methylene Blue (MB) Staining of Tissues for Histology, Micro-Dissection and RNA Isolation

*Tissue de-paraffinization and re-hydration:* All tissues were processed according to our standard methodology: FFPE 10 µm tissue cuts were first treated at 70 °C for 1 h in a dry oven to facilitate paraffin melting. Tissue cuts were then incubated for 10 min at RT in a sequential order with the following solutions for further de-paraffinization and rehydration: (1) 3 × Roti-Histol (Carl Roth, #6640.4); (2) 100% Ethanol (Carl Roth, #5054.3); (3) 96% Ethanol; (4) 70% Ethanol and (5) dH_2_O with 0.1% DEPC. Matched FF 10 µm tumor tissues were incubated 2 times for 5 min with 70% Ethanol and dH_2_O to remove Tissue-Tek^®^.

*MB tissue staining:* Following tissue de-paraffinization and re-hydration, tissue cuts were incubated with or without MB (1% solution, Carl Roth #AE64.1) for: (1) FFPE and FF 10 µm tissue sections on glass slides to demonstrate specific histological staining and tumor quantity (Figure 1); or (2) RNA isolation from microdissected MB stained tissue regions. For staining, tissues were incubated for 20 s in MB 1:3 diluted in DEPC treated tap-H_2_O (the time was empirically determined in order not to saturate the tissue) and washed 2 times each 10 s with sterile dH_2_O (total time ~1 min). Then samples were centrifuged (16,000× *g* for 5 min at RT) to remove the dH_2_O, and the first solution of the corresponding RNA method was added.

*Hematoxylin/Eosin (H&E) tissue staining:* A modified shortened protocol for H&E histological staining of tissue cuts was used: 1 mL hematoxylin (Sigma Aldrich, Taufkirchen, Germany #GHS332) was added to the slide and incubated for 3 min at RT. The slides were washed 2 times with dH_2_O (each 1 min) and 1 mL of bluing buffer (Dako #CS702) for 1 min was added. After 1 time washing with dH_2_O (1 min), 1 mL eosin was added for 1 min at RT, again washed with dH_2_O (total time ~10 min). Lastly, all H&E and MB stained tissue slides for histological analyses were cover slipped with glass slides and a mounting buffer (Eukitt, Orsa-tec, #6.00.01.0001.06.01.01).

### 4.4. RNA Isolation from FFPE and Tissue-Tek^®^ FF Matched Tumor Tissue Samples Using Different Methodologies

Six MIBC FFPE and matched FF tumor tissues (*n* = 6), each 10 µm thick, were stained either with or without MB as above and then immediately processed using different RNA isolation methods to determine RNA integrity. Regarding all methodologies below, the Proteinase K digestion step was extended to overnight for all FF tissues, if applicable. The four RNA isolation methods are shown in Table 1 and explained below with some modifications.

The following total RNA isolation methodologies were used for comparison: **(****1)** Maxwell RSC DNA Blood kit (Promega, #AS1400) (Method A). For RNA tissue extraction a purchased incubation buffer (Promega, #D920B) was added replacing the DNA extraction buffer. Briefly, samples were heated to 80 °C for 10 min, then Proteinase K and Protector RNase Inhibitor (Roche, #3335399001) were added, followed by incubation at 56 °C for 1 h while shaking. Afterwards, an RNA lysis buffer (Promega, #Z3051) was added and the samples were pipetted into the cartridges (supplied by the manufacturer) and processed by a Maxwell RSC machine. Since this isolation method does not involve a DNase I step, 20 U RNAse-free DNase I (Roche, #04716728001) was added for 1 h at 37 °C. **(****2)** Maxwell RSC RNA FFPE kit (Promega, #AS1440) (Method B). Samples were processed according to manufacturer’s instructions, starting at the Proteinase K incubation along with the addition of an RNA Protector RNase Inhibitor. **(****3)** High Pure FFPE RNA Isolation Kit (Roche, #06650775001) (Method C). Samples were processed according to manufacturer’s instructions, starting at the Proteinase K incubation along with the addition of an RNA Protector RNase Inhibitor. **(4)** XTRAKT FFPE kit (STRATIFYER Molecular Pathology GmbH, Cologne, Germany) (Method D). Samples were processed according to manufacturer’s protocol. **(****5)** Trizol (Thermo, #15596-026) (Method E). Only FF MIBC tissues and FF normal placental tissues (first treated for 5 s using a Micro-Dismembrator (853162, B. Braun) were isolated for RNA according to manufacturer’s instructions with some modifications. Briefly, Isopropanol was added to the aqueous phase (1:1, *v*/*v*) in the presence of 20 µg Glycogen (Sigma-Aldrich, #10901393001) at −20 °C overnight, the samples were centrifuged (16,000× *g*, 1 h, 4 °C) and the supernatant removed. RNA was washed with 80% ethanol, air dried, solubilized in DEPC treated dH_2_O, DNAse I treated as above, precipitated with 0.6 M sodium acetate in the presence of ethanol and incubated overnight at −20 °C. Samples were centrifuged, washed with 80% ethanol, air dried and solubilized with DEPC treated dH_2_O.

### 4.5. Purification and Quality Control of RNA from FFPE and FF Tissue Samples

After RNA isolation was performed using the different methods described above, all RNAs except for Trizol isolated, were further purified using an equal volume of chloroform (Fisher-Sci., #366919), centrifuged (16,000× g, 10 min, 4 °C), a back-extraction with TE buffer (10 mM Tris pH 7.6, 1 mM EDTA pH 8.0) to ensure no loss of RNA during extraction and then the aqueous phases containing RNA were precipitated with 100% ethanol in the presence of Glycogen and 0.6 M sodium acetate and incubated at −20 °C overnight. RNA samples were centrifuged (16,000× *g*, 1 h, 4 °C) and the supernatant removed. RNAs were washed with 80% ethanol, air dried and solubilized in DEPC treated dH_2_O and then quantified using the QuantiFluor^®^ RNA System (Promega, #E3310) and Nanodrop 2000 (Thermo Scientific). RNA quality, like the RNA Integrity Number (RIN) and the DV200 was assessed using the Bioanalyzer 2100 with the corresponding 2100 Expert Software from Agilent.

### 4.6. cDNA Synthesis and qPCR

For cDNA a total of 1 µg RNA per sample was reverse transcribed to obtain a concentration of 20 ng/µL cDNA using Applied Biosystems High Capacity cDNA Reverse Transcription Kit (ABI, #4368813). For qPCR, 40 ng cDNA, SYBR Select Master Mix (ABI, #4472919), 200 nM of each primer were analyzed using the StepOnePlus Real-Time PCR Systems (Applied Biosystems^TM^). All primer nucleotide sequences for qPCR are shown in Appendix A.

### 4.7. The nCounter™ System: Multiplex RNA Hybridization

Thirty-four reference genes and six MIBC subtyping target genes with NCBI Accession numbers (Appendix A) were analyzed as nCounter™ probes. The nCounter™ assay was performed according to the manufacturer’s recommendations within a range from 100 to 500 ng of total RNA. Raw data were then pre-processed using the nSolver 4.0 software and normalized to six synthetic internal positive controls (POS A, POS F) to measure the hybridization efficiency.

### 4.8. Whole Transcriptome mRNA Sequencing

mRNA sequencing was performed using the Lexogen QuantSeq^TM^ 3′ mRNA-Seq Kit FWD (Lexogen GmbH) which allows for the generation of Illumina-compatible libraries from poly-adenylated RNA. 250 ng of purified total RNA were used in input for library preparation which was carried out according to manufacturer’s instructions. The QuantSeq protocol creates libraries close to the 3′ end of the transcripts generating only one fragment per transcript. Sequencing of the QuantSeq libraries was performed on the Illumina NovaSeq 6000 platform (single end; 1 × 75 bp) yielding at least 20 M clusters per sample. Pre-processing of demultiplexed raw data (FASTQ files) was performed via the nf-core RNA-seq pipeline v.3.3 [57]. In particular, reads were adapter- and quality-trimmed using Trim Galore v.0.6.6 (Babraham, Bioinformatics) and mapped to the Ensembl human genome assembly GRCh38 (release 104) [58] using STAR v.2.6.1d [59]. Read counting at transcript level was performed using Salmon v.1.4.0 [60] relying on the Ensembl gene annotation file release 104. Gene-level counts were generated from Salmon’s transcript-level quantification files relying on the R-package tximport v.1.16 [61] within R v.4.0.3 [62].

### 4.9. MIBC Subtyping

Twenty eight MIBC samples were classified into luminal and basal subtypes based on three different approaches: (1) Protein immunohistochemistry (IHC) determination of luminal and basal subtypes adapted from a previous report [63]. (2) RNA subtyping using multiplex RNA hybridization (Nanostring) gene expression data (Appendix A). Data were normalized to the best reference genes determined in this study, log2 transformed and then subtypes were identified by unsupervised hierarchical clustering (average linkage algorithm with Euclidean distance as metric scale); (3) RNA reference subtyping using mRNA sequencing data. Gene counts were log2 transformed and then MIBC consensus subtypes calling was performed in R v. 4.1.0 using the single sample classifier R-package BLCAsubtyping v.2.1 (https://github.com/cit-bioinfo/BLCAsubtyping) [25].

### 4.10. Statistical Analyses

All statistical analyses such as two-tailed Mann––Whitney U test, coefficient of variation (COV), ANOVA and Pearson correlation coefficients (R^2^) as well as all graphs, were performed using GraphPad Prism 9. A *p*-value < 0.05 was considered statistically significant. Unsupervised hierarchical clustering was performed with JMP SAS 13.2.

## 5. Conclusions

In conclusion, we outline several messages when isolating RNA from archival tissues. MB can easily be used for staining tissues of interest for microdissection and RNA isolation resulting in no artifacts due to its chemical properties. We recommend employing MB when extracting RNA from FF tissues especially using Methods B and C, which leads to improved RNA integrity and reference gene expression, like *18SrRNA* and *5SrRNA* and *5.8SrRNA*. When isolating RNA from more commonly available FFPE archival tissues, implementing MB becomes a choice depending on the amount of targeted tissue within a sample for RNA isolation and analyses. This becomes important since most methods extracting RNA from FFPE tissues did not show any differences in RNA integrity with or without MB. However, when using microdissection, MB helps to identify specific tissue regions leading to enrichment of an RNA population. As a recommendation, staining with MB and implementing Methods B or C results in excellent *18SrRNA* using qPCR, which also demonstrated 100% congruence of MIBC tumor subtyping gene expression using direct multiplex RNA hybridization compared with IHC and mRNA sequencing. Our above recommendations could also be important for spatial sequencing and laser dissection.

## Figures and Tables

**Figure 1 ijms-23-10267-f001:**
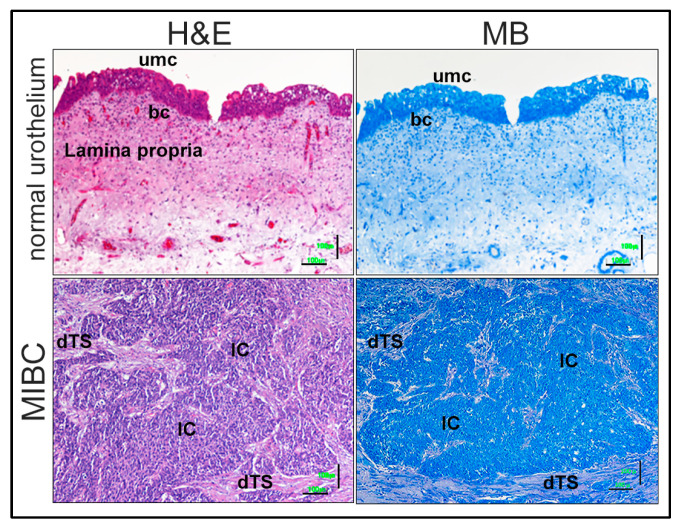
Example of consecutive FFPE tissue sections showing state of the art hematoxylin and eosin (H&E) and similar methylene blue (MB) staining of normal urothelium and muscle invasive bladder cancer (MIBC). Top microscopy photos represent normal urothelium H&E staining (left) and MB (right). Bottom photos represent MIBC H&E (left) and MB tissue stains (right). Different histological tissue components: surface epithelia cells: umc = umbrella cells; bc = basal cells; Lamina propria; dTS = desmoplastic tumor stroma; IC = invasive carcinoma. Lower right bar = 100 µm.

**Figure 2 ijms-23-10267-f002:**
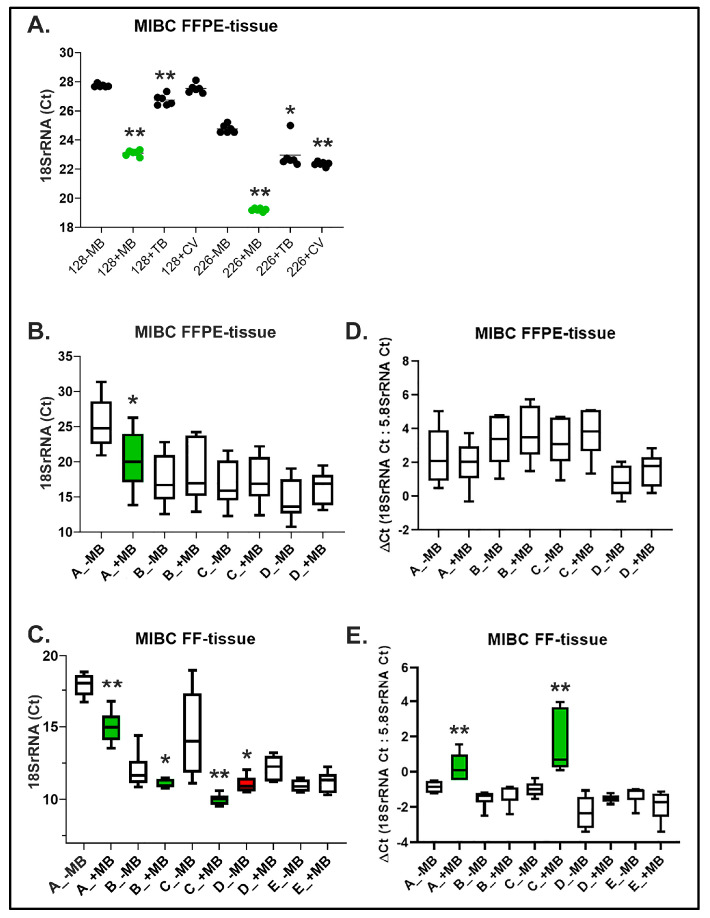
MIBC qPCR *18SrRNA* values of RNA isolated from FFPE and FF tissues using different isolation methods. (**A**) Graph shows *18SrRNA* Ct values (*Y*-axis) from RNA isolated from two FFPE MIBC tissues (#128, #226) using Method A (Table 1). MIBC tissues were microdissected from Methylene blue (MB); Toluidine blue (TB) or Crystal violet (CV) stained samples prior to RNA extraction. Graph shows tissue 128 without (−) MB compared to 128 with (+) MB (green dots) ** *p* = 0.0022 and 128 +TB: ** *p* = 0.0022; and tissue 226 −MB compared to 226 +MB (green dots): ** *p* = 0.0022 and 226 +TB: * *p* = 0.0411 and 226 +CV: ** *p* = 0.0022, *n* = 6 technical replicates. (**B**) Each box graph shows *18SrRNA* Ct values (*Y*-axis) from RNA isolated from six different FFPE MIBC tissues with (+) or without (−) MB prior staining using four different isolation methods (A–D) (Table 1); *n* = 6 technical replicates; * *p* = 0.0481 Method A (graph in green). All other results either with (+) or without (−) MB were not statistically significant. (**C**) Each box graph shows *18SrRNA* Ct values (*Y*-axis) from RNA isolated from the six different FF patient matched MIBC tissues to FFPE tissues (see **B**) above with (+) or without (−) MB prior staining using five different isolation methods A–E (Table 1); A −MB to A +MB (green graph), ** *p* = 0.0043, B −MB to B +MB (green graph), * *p* = 0.039, C −MB to C +MB (green graph), ** *p* = 0.0022, D −MB to D +MB (red graph), * *p* = 0.026; all comparisons to A +MB with B +MB, C +MB, D +MB and E +MB were ** *p* = 0.0022; all were performed with *n* = 6 technical replicates. (**D**) Box graphs for isolated RNA from MIBC FFPE tissues after normalization of *18SrRNA* with *5.8SrRNA* (ΔCt). (**E**) Box graphs for isolated RNA from MIBC FF tissues after normalization of *18SrRNA* with *5.8SrRNA* (ΔCt). ** *p* = 0.0065 (A +MB) and ** *p* = 0.0022 (C +MB).

**Figure 3 ijms-23-10267-f003:**
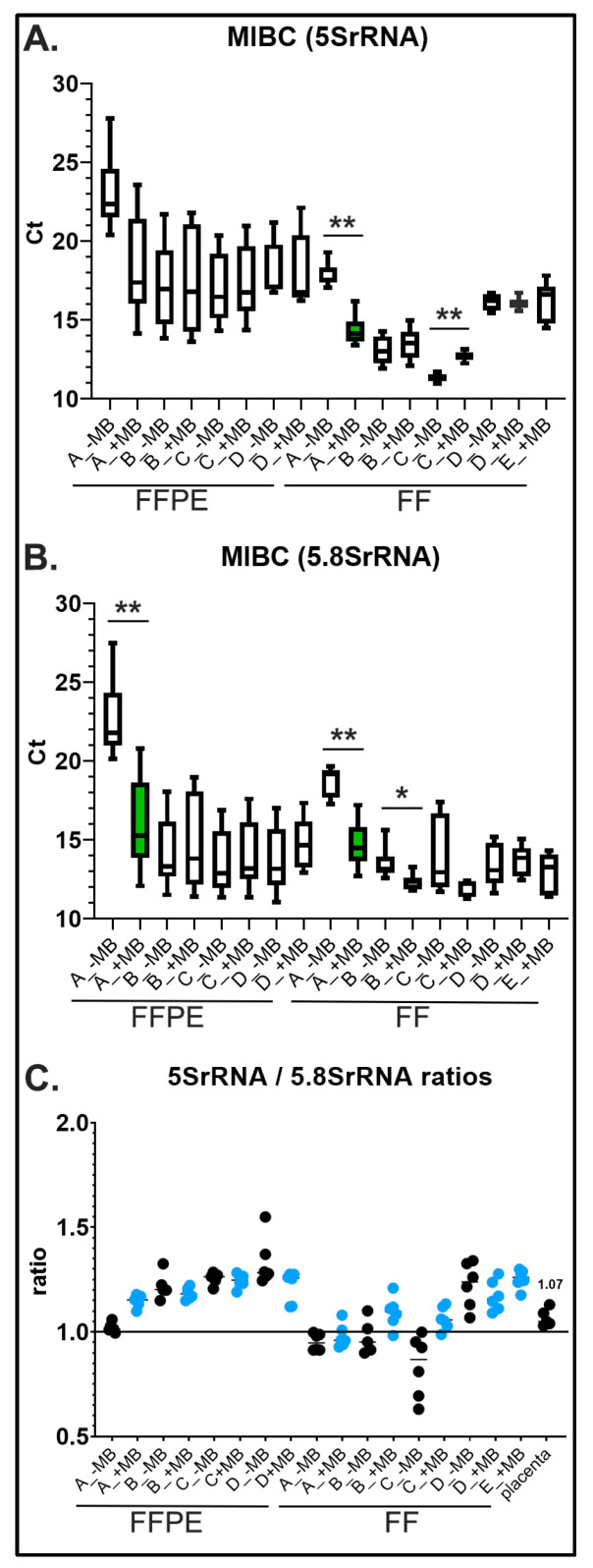
MIBC qPCR *5SrRNA* and *5.8SrRNA* values from RNA isolated from FFPE and FF tissues using different isolation methods. (**A**) Each box graph shows *5SrRNA* Ct values (*Y*-axis) from RNA isolated from six different FFPE MIBC tissues and patient matched FF MIBC tissues with (+) or without (−) MB prior staining using four different isolation methods (A–D) (Table 1); ** *p* = 0.0022, *n* = 6 technical replicates. (**B**) Each box graph shows *5.8SrRNA* Ct values (*Y*-axis) from RNA isolated from six different FFPE MIBC tissues and patient matched FF MIBC tissues with (+) or without (−) MB prior staining using four different isolation methods for FFPE (A–D) and five isolation methods for FF (Table 1); ** *p* = 0.0043; * *p* = 0.026, *n* = 6 technical replicates. (**C**) Graph shows ratios (*Y*-axis) of *5SrRNA*/*5.8SrRNA* Ct values from RNA isolated from six different FFPE MIBC tissues and patient matched FF MIBC tissues with (+) or without (−) MB prior staining using four different isolation methods for FFPE (A–D) and five isolation methods for FF tissues (Table 1). Far right represents control human placenta (*n* = 5) (flash frozen tissue in liquid nitrogen and isolated with Method E) used for comparison.

**Figure 4 ijms-23-10267-f004:**
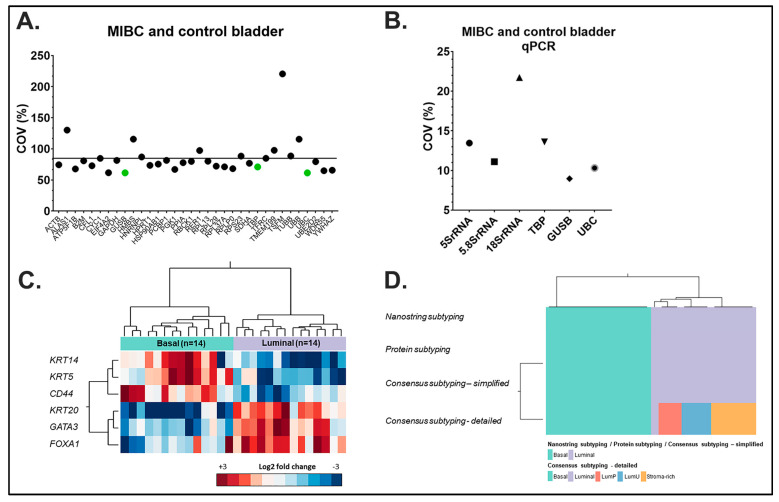
Identification of stable reference genes for MIBC and control tissues. (**A**) Using direct multiplex RNA-hybridization of MIBC (*n* = 28) and control bladder (*n* = 5), the coefficient of variation (COV) in % was calculated. Line shows mean COV of all reference genes and green points represent the most stable expressed reference genes. (**B**) Graph (*Y*-axis) demonstrates the COV % of six different reference genes using qPCR: *5SrRNA* (sharp circle), *5.8SrRNA* (square), *18SrRNA* (triangle) and confirmation of the three most stable reference genes, *TBP* (triangle), *GUSB* (diamond) *UBC* (blurred circle) for combined MIBC and control bladders. Although *18SrRNA* showed a relative high COV of 21.7%, it represented the gene with the highest detection rate (lowest Cts). (**C**) Unsupervised hierarchical cluster analysis of 28 MIBC based on six subtyping genes (basal differentiation: *KRT5*, *KRT14*, *CD44*; luminal differentiation: *KRT20*, *GATA3*, *FOXA1*) assessed by multiplex RNA-hybridization and normalized according to the three optimal housekeeping genes *GUSB*, *TBP* and *UBC*. Fourteen samples were classified each as basal or luminal. (**D**) Comparison of multiplex RNA-hybridization subtyping, protein IHC subtyping (derived by IHC assessment of KRT5/CK5, KRT14/CK14, CD44, KRT20/CK20, GATA3 and FOXA1) and transcriptome consensus subtyping. Consensus subtyping is displayed in two categories: (1) “simplified” divided into a luminal and a basal category and (2) “detailed” divided into a basal and four luminal categories (luminal, luminal papillary/LumP, luminal unstable/LumU and stroma-rich). Importantly, concordance rates between multiplex RNA-hybridization subtyping and other approaches amounted to 100%; furthermore, all luminal subsets were correctly classified as luminal by multiplex RNA-hybridization subtyping.

**Table 1 ijms-23-10267-t001:** RNA isolation methods summary.

	A		B		C		D		E
Company	Promega		Promega		Roche		Stratifyer		(Trizol, Ambion)
Order #	AS1400		AS1440		6650775001		XTRAKT FFPE		15596-026
Column chromatography	no		no		yes glass fiber fleece		no		no *^1^
Magnetic beads	yes cellulose coated		yes silica coated		no		yes germanium coated		no *^1^
tissues	FFPE	FF	FFPE	FF	FFPE	FF	FFPE	FF	FF
Cost [€] *^2^	340	340 + extra components	340		304		580		160
# of reactions	48		48		50		96		125 *^3^
Cost per reaction [€]	7.08	7.92	7.08		6.08		6.04		1.28
Total time *^4^	3 h		3.5 h		2.5 h (+1x O.N.)		3.5 h		3 h (+2x O.N.)
special equipment	Maxwell 16LEV or RSC48		Maxwell 16LEV or RSC48		--		magnetic separation rack		--
RIN + MB *^5^	2.3 3.1		2.3 2.4		2.3 2.5		2.2 2.3		2.4
RIN − MB *^5^	2.3 3.1		2.3 2.4		2.3 2.5		2.2 2.3		2.4
DV200 + MB [%] *^6^	~60	~70	~45	~55	~60	~60	~50	~50	~35
DV200 − MB [%] *^6^	~50	~70	~50	~70	~60	~45	~65	~60	~35

*^1^ = phenol based organic solvent; *^2^ = net for Germany; *^3^ = calculated; *^4^ = total hands-on time; *^5^ = Mean RIN; *^6^ = Mean DV200. No statistical significance between the different isolation methods for FFPE and FF was found performing an ANOVA analyses for the RIN and DV200.

## Data Availability

The datasets used and/or analyzed during the current study are available from the corresponding author on reasonable request.

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
