# Peer review of "Improved Bladder Tumor RNA Isolation from Archived Tissues Using Methylene Blue for Normalization, Multiplex RNA Hybridization, Sequencing and Subtyping"

_ijms, 2022, doi:10.3390/ijms231810267_

Round 1

Reviewer 1 Report

This is a well-written article with clearly presented results by the authors and follows a natural progression. 

1. The authors mention the improvement of RNA integrity and reference gene expression. Does this imply the mentioned gene expressions such as 18SrRNA and 590 5SrRNA and 5.8SrRNA? 2.  page 476: What is the rationale behind the short incubation time of 20 seconds? 3.  As the authors have mentioned an improved RNA isolation from the archived tissues using MB, can you please add a brief section summarizing the other methods and then make a comparison based on the statistical analysis? 4. Can you provide the relationship between one or more independent variables and a response, dependent, or target variable? Maybe by using a regression model.

Author Response

Reviewer 1:

This is a well-written article with clearly presented results by the authors and follows a natural progression. 

  1. The authors mention the improvement of RNA integrity and reference gene expression. Does this imply the mentioned gene expressions such as 18SrRNA and 590 5SrRNA and 5.8SrRNA? 

Author’s reply: First, we thank the reviewer for the positive feed-back and an opportunity to answer questions regarding our manuscript.

We focused on muscle invasive bladder cancer in this investigation. For all real time PCR analyses regarding the RNA integrity following with or without MB treatment, we tested 18SrRNA, 5SrRNA and 5.8SrRNA.

  1. page 476: What is the rationale behind the short incubation time of 20 seconds? 

Author’s reply: We apologize for not explaining this point further. The 20 seconds for tissue staining were evaluated empirically and was found to be the most optimal time-frame for histological staining. Longer than 20 seconds MB treatments resulted in a more saturated appearance, thus reducing histological details (see sentence in Materials and Methods section 4.3).

  1. As the authors have mentioned an improved RNA isolation from the archived tissues using MB, can you please add a brief section summarizing the other methods and then make a comparison based on the statistical analysis? 

Author’s reply: Thank you for the suggestion to compare all methodologies and we agree this is important for all readers. We have now added a new paragraph summarizing the general characteristics of all five methodologies performed in our study (Results section 2.2). We also expanded Table 1. In addition to our statistical analyses for reference genes comparing matched tissues isolated from MB or no treatment (Figures 2,3), we also analyzed RIN and DV200 values (new Figure S2 A-E) from the different isolation methods (FFPE and FF tissues with or without MB) using analyses of variation (ANOVA) and did not find any statistical differences. This is now stated in the Table 1 legend.

  1. Can you provide the relationship between one or more independent variables and a response, dependent, or target variable? Maybe by using a regression model.

Author’s reply: This is a very interesting and important suggestion. However, due to the non-significance of the RIN and DV200 variables for the different isolation methods (Table 1 legend), which could have resulted in the most important changes, we did not proceed with a complete regression model with all variables. We think that: 1) the differences of the stated variables are not substantial enough to yield significances or 2) the number of analyzed tissues (n= 6) was not high enough, even though 5 different isolation methods were used. Our main point to address in our study was to determine if MB has an impact on the RNA integrity during isolation from FFPE and FF tissues. The significant differences seen for Method A and C in the presence of MB is based from an analyses of RNA integrity via reference genes, which indicates that the isolation methods A and C with MB are advantageous above others and without MB.

Reviewer 2 Report

First at all I’m very surprised by the historical flavour of the Introduction section. While I highly value to read on the historical origins of molecular biology I think that this approach is more for a revision manuscript than for an original paper presenting results. Since this work compares five different technologies for RNA isolation I would find more informative a comparative description of these techniques in the Intro section.

Major concerns.

-It is very important that authors explain better the differences among the RNA isolation methods used. What are the differences at the tissue processing step, and at the lyses, column and post column processing steps?  Table 1 should be improved to include the requested data.

-Figure 1, panel 4 (MIBC/MB) there’s no magnification bar and furthermore it seems to be at a different magnification than panel 3. Furthermore, I’m not so sure that MB staining is superior to H&E since there are a number of structures at the basal region of the “lamina propia” that are positively stained with H&E but not with MB.

-Figure 2 shows direct Ct values for 18S RNA. I am sorry, but without any further normalization these values are difficult to interpretrate since they may just reflect the amount of tissue loaded onto the PCR, even if these have been carefully measured. The gold standard for PCR indicates that authors SHOULD normalize these Ct values with another transcript (delta-Ct). In this sense, it is striking the very high variability in Figure 2, panel C, and sample C-MB. How do authors explain this variability? It is striking that this can also be seen in Figure 3, panels B and C where the sample C-MB is clearly an outlayer

- Since RNA degradation is a point especially when dealing with stored tumor samples, it would be interesting to show the measures of RNA integrity in a new table and better to show a visual description (chromatogram, agarose gel picture) of DNAased samples to get a more  complete picture of the topic.

Author Response

Reviewer 2:

First at all I’m very surprised by the historical flavour of the Introduction section. While I highly value to read on the historical origins of molecular biology I think that this approach is more for a revision manuscript than for an original paper presenting results. Since this work compares five different technologies for RNA isolation I would find more informative a comparative description of these techniques in the Intro section.

Author’s reply: We thank the reviewer for helping us to streamline our manuscript further. We agree with the reviewers’ opinion not to write in a review style and that we were a bit over ambitious about referring to the history of dyes and contribution to different fields of research. We now have cut two historical references (refs. 11 and 12) and part of a sentence regarding MB patenting and MB for chromosomal banding (ref. 20). However, we want to include some background of MB, since it is implemented for a variety of both clinical and research applications for over 100 years. We also have now added a new paragraph summarizing the general characteristics of all five methodologies performed in our study (Results section 2.2). We also expanded Table 1.

Major concerns.

-It is very important that authors explain better the differences among the RNA isolation methods used. What are the differences at the tissue processing step, and at the lyses, column and post column processing steps?  Table 1 should be improved to include the requested data.

Author’s reply: We thank the reviewer and agree it is very important to have all information summarized for the readers. The general differences between the five RNA isolation methods are now summarized in the new paragraph (Results section 2.2) and also in our newly improved and expanded Table 1.

-Figure 1, panel 4 (MIBC/MB) there’s no magnification bar and furthermore it seems to be at a different magnification than panel 3. Furthermore, I’m not so sure that MB staining is superior to H&E since there are a number of structures at the basal region of the “lamina propia” that are positively stained with H&E but not with MB.

Author’s reply: We apologize for the missing magnification bar in Figure 1. We re-photographed Fig. 1 MIBC/MB now with bars, which is the same original magnification than before. We agree with the reviewer concerning H&E. Our goal was not to make a direct comparison between MB and H&E and determine which stain is better. H&E is the state of the art staining for pathological diagnosis defining detailed histology. We wanted to stress that MB can easily be used for histological staining for microdissecting important tissue areas for RNA isolation (tumor vs control urothelium). Lastly, we have now toned down our description of the stains in Results section 2.1 and state that MB stained tissues are similar to H&E staining regarding histological appearance in Figure 1. In the Discussion (page 12) and Materials and Methods (section 4.3) we state that MB is similar in staining and has advantages over H&E regarding staining time (MB 1 minute total time with washes vs a shortened H&E protocol 10 minutes with washes), and costs, which could be important for RNA isolation, especially regarding RNAses, for tissue microdissection, laser capture microdissection or spatial sequencing.

-Figure 2 shows direct Ct values for 18S RNA. I am sorry, but without any further normalization these values are difficult to interpretrate since they may just reflect the amount of tissue loaded onto the PCR, even if these have been carefully measured. The gold standard for PCR indicates that authors SHOULD normalize these Ct values with another transcript (delta-Ct). In this sense, it is striking the very high variability in Figure 2, panel C, and sample C-MB. How do authors explain this variability? It is striking that this can also be seen in Figure 3, panels B and C where the sample C-MB is clearly an outlayer.

Author’s reply: We absolutely agree with the reviewer in that no “target” gene can be evaluated without normalization to reference genes. However, in the present circumstance, we used a common and established reference gene (18SrRNA) for RNA analysis by calculating and presenting the different Cts. We also implemented the 5SrRNA and 5.8SrRNA reference genes as well. Importantly as asked by the reviewer we re-analyzed all data and now additionally present the Delta-Cts (normalizing 18SrRNA with 5.8SrRNA transcripts) of the different FFPE and FF isolations (new Fig. 2 D,E). In the results on page 7 we included a new paragraph to explain the new Figure 2 D, E:  “Although 18SrRNA is a well-known reference gene, importantly we normalized to the total RNA based from several quantification methods including a fluorescent RNA stain detecting down to 5 ng/ml RNA [35], and additionally normalized 18SrRNA to another reference gene (5.8SrRNA) to calculate deltaCts. Analysing deltaCts of the RNAs isolated from the different methods with MB showed that the highly significant changes (A +MB and C +MB) did not change in significance (Figure 2E). However, the lower significant changes using 18SrRNA Ct (A +MB FFPE, B +MB FF and D -MB FF) were no longer significant (Figure 2B-E).

To address the value of calculating Cts or deltaCts especially using rRNAs as references and to address RNA integrity one must follow specific guidelines from the literature. For example, we added the sentence in the Introduction on page 3: “Considering the recommendation that rRNAs, which constitute 85-90 % of total cellular RNA are highly useful and accurate internal controls and more reliable than other reference genes, a normalization of the isolated RNA to total RNA using especially fluorescent RNA stains is advised [30]“.  We used these guidelines as shown in Material and Methods (sections 4.4 and 4.5); Results (section 2.2).  To highlight our RNA quantification methods (Materials and Methods 4.5) we performed QuantiFluor® RNA System (Promega, #E3310) and Nanodrop 2000 (Thermo Scientific). RNA quality, like the RNA Integrity Number (RIN) and the DV200 was also assessed using the Bioanalyzer 2100 with the corresponding 2100 Expert Software from Agilent (new Figure S2). We hope that the reviewer now agrees that our strategy and accuracy for RNA quantification are reputable.

Another point we would like to stress is the power of rRNAs as reference genes alone and their use for indicating RNA integrity when analyzing samples. This is exemplified by calculating Cts to screen for especially RNA degradation among tissue samples. As one example from the literature by Gilli F and van Beers M et al (J of Immunological Methods, 2008) the authors demonstrated that “levels of damaged cells were easily evaluated based on the threshold cycle (Ct) values of the housekeeping gene 18SrRNA”. In addition, all authors’ results were based upon extraction of total RNA.

Furthermore, in order to answer the question, if different RNAs using an isolation method are comparable, we further produced standard curves of two different RNAs isolated with Method B plus MB for the reference genes 18SrRNA, TBP and UBC (new suppl. Fig. 2 F-H). Both RNAs showed highly similar standard curves, slopes and R2 (Pearson correlation coefficients), which confirm highly similar amplification efficiencies and linearity (see Results p. 11, section 2.5). Thus, the Cts of reference genes per se are comparable, but we definitely agree with the reviewer that “target genes” need to be normalized. We added the following sentence after a description of the slopes (Figure S2F-H) on page 11 in the Results: “Interestingly, comparing the standard curves, slopes and R2 of RNA isolated from two different MIBC FF tissues further support that the isolation technique results in similar RNA concentrations and degradations and confirms highly similar amplification efficiencies and linearity.”

Regarding the last reviewers point in this question, concerning the variability (higher standard deviations) of C -MB 18SrRNA for FF (Fig. 2C) and C -MB 5.8SrRNA for FF (Fig. 3B), it is not seen with C -MB 5SrRNA for FF (Fig. 3A). The higher standard deviations could be due to the more variable RNA qualities for Method C without MB (in the presence of MB there were lower variabilities) as well as which reference gene is used. This data is further described on page 11 in the Results and a new Figure S2 F-H: “(Figure S2F-H). Again, comparing the standard curves and slopes of RNA isolated from two different MIBC FF tissues further supports that the isolation technique results in similar RNA concentrations and degradations”.

- Since RNA degradation is a point especially when dealing with stored tumor samples, it would be interesting to show the measures of RNA integrity in a new table and better to show a visual description (chromatogram, agarose gel picture) of DNAased samples to get a more  complete picture of the topic.

Author’s reply: We thank the reviewer for this comment. We have now added all RIN and DV200 values to the new Table 1 and performed statistics between the different samples per methodology. We added a new Supplement Figure 2 (Figure S2) showing the chromatograms with RIN and DV200 values.  

Round 2

Reviewer 1 Report

This is a well-written article with clearly presented results by the authors and follows a natural progression. Thank you for addressing the questions presented in the initial review.

Reviewer 2 Report

The authors have improved the manuscript as requested